# Embryonic Trophectoderm Secretomics Reveals Chemotactic Migration and Intercellular Communication of Endometrial and Circulating MSCs in Embryonic Implantation

**DOI:** 10.3390/ijms22115638

**Published:** 2021-05-26

**Authors:** Alexandra Calle, Víctor Toribio, María Yáñez-Mó, Miguel Ángel Ramírez

**Affiliations:** 1Departamento de Reproducción Animal, Instituto Nacional de Investigación y Tecnología Agraria y Alimentaria (INIA), 28040 Madrid, Spain; calle.alexandra@inia.es; 2CBM-SO, Instituto de Investigación Sanitaria Princesa (IIS-IP), 28049 Madrid, Spain; victor.toribio@uam.es (V.T.); maria.yannez@uam.es (M.Y.-M.); 3Departamento de Biología Molecular, UAM, 28049 Madrid, Spain

**Keywords:** mesenchymal stromal cells, trophectoderm, epithelial to mesenchymal transition, cell migration, extracellular vesicles

## Abstract

Embryonic implantation is a key step in the establishment of pregnancy. In the present work, we have carried out an in-depth proteomic analysis of the secretome (extracellular vesicles and soluble proteins) of two bovine blastocysts embryonic trophectoderm primary cultures (BBT), confirming different epithelial–mesenchymal transition stages in these cells. BBT-secretomes contain early pregnancy-related proteins and angiogenic proteins both as cargo in EVs and the soluble fraction. We have demonstrated the functional transfer of protein-containing secretome between embryonic trophectoderm and maternal MSC in vitro using two BBT primary cultures eight endometrial MSC (eMSC) and five peripheral blood MSC (pbMSC) lines. We observed that eMSC and pbMSC chemotax to both the soluble fraction and EVs of the BBT secretome. In addition, in a complementary direction, we found that the pattern of expression of implantation proteins in BBT-EVs changes depending on: (i) their epithelial–mesenchymal phenotype; (ii) as a result of the uptake of eMSC- or pbMSC-EV previously stimulated or not with embryonic signals (IFN-τ); (iii) because of the stimulation with the endometrial cytokines present in the uterine fluid in the peri-implantation period.

## 1. Introduction

Embryo implantation is a crucial step for pregnancy establishment. In ruminants, noninvasive trophoblast attaches and adheres to the uterine endometrium, followed by the formation of the placenta. It has been demonstrated that, during the adhesion of the embryo to the endometrium on day 22, the embryonic trophectoderm upregulates the expression of genes characteristic of an epithelial to mesenchymal transition (EMT) [1]. Lee et al. demonstrated that cell–cell interaction continues during conceptus implantation into the endometrium and that EMT is the key process for many events that facilitate embryonic development, for which the expansion of epidermal basal compartments is required [2]. During this period, trophectoderm cells must be more flexible, enabling the formation of binucleated and trinucleated cells. It was found that binucleated trophoblast cells exhibit intermediate characteristics between epithelial and mesenchymal phenotypes [3]. Although EMT occurs in different biological processes, the first type of EMT is thus associated with implantation, embryo formation, organ development, and the generation of a variety of cell types with mesenchymal phenotypes [4].

Although this EMT process has been reported in bovine trophectoderm cells during embryo implantation [1,5,6], its consequences on the cell secretome have not yet been studied. Embryo–maternal communication through secretome, including both extracellular vesicles and soluble proteins, is essential for successful embryonic implantation.

To date, most of the studies that aim to investigate the process of bovine embryo–maternal communication [1,5,6] have been performed in vitro with a trophectoderm primary culture (CT-1) established by cocultivation with mouse feeder layers [7,8,9]. Our group isolated and characterized eight primary cultures of bovine blastocyst embryonic trophectoderm cells, (BBT, through a novel biopsy and culture system without the need to use coculture with murine cells [10]). The different trophoblastic primary cultures established showed the expression of genes from early trophoblastic markers (*CDX2*, *ELF5*), mononucleated cells (*IFNT*), and binucleated cells (*PAG1*, *PRP1*, and *CSH2*), which varied with time in culture, indicating that these primary cultures are dynamic populations [10].

Recently, our group described, for the first time, the isolation, immortalization, and characterization of endometrial mesenchymal stem cell lines (eMSC) from different estrous cycle stages, with a clear mesenchymal pattern and immunomodulatory properties. Our study also reported that the migratory capacity of these eMSCs was increased towards an inflammatory niche but was reduced in response to the presence of the bovine implantation cytokine interferon tau-1 (IFN-τ). The combination of both signals (pro-inflammatory and implantation) would ensure the retention of eMSC in the case of pregnancy. Interestingly, in the absence of embryo stimuli, eMSC showed an apparent mesenchymal to epithelial transition stage [11]. Additionally, we have just reported the successful isolation for the first time in bovines, of peripheral blood MSC (pbMSCs) lines with classical MSC markers, multipotent capacity, and immune-suppression activity. Interestingly, in contrast to MSCs derived from endometrial tissue, some pbMSC lines showed chemotactic activity towards the IFN-τ implantation cytokine as well as towards a raw secretome of BBT. Our results would thus suggest that circulating MSCs are present in the peripheral blood under healthy conditions and can be actively recruited to the implantation niche during pregnancy. Retention of eMSC and recruitment of pbMSC would ensure the necessary immunorepression to prevent embryo rejection by the maternal organism [12].

The main objective of the present work is to carry out an in-depth analysis of the secretome (extracellular vesicles and soluble proteins separately) of two BBT primary cultures that represent two different stages of epithelial–mesenchymal transition of the embryonic trophectoderm, as well as their effects on chemotactic migration of maternal eMSC and pbMSC. To demonstrate the functional consequences of this secretome-based maternal–embryonic communication, we have analyzed both the uptake of EVs from BBT by maternal MSC as well as the changes in BBT secretome as a result of the uptake of eMSC- or pbMSC-EVs or after treatment with endometrial cytokines present in the uterine fluid in the peri-implantation period.

## 2. Results

### 2.1. The Proteomic Profiles of BBT-9 and BBT-18 Secretomes (EVs and Soluble Proteins) Are Compatible with Different EMT Stages

To analyze the components of the BBT secretome, conditioned media from cell cultures of BBT-9 and BBT-18 were fractionated by SEC. Collected fractions were quantified with a Nanodrop spectrophotometer, characterized by bead-assisted flow cytometry using antibodies against classical EV markers, CD63 and CD9 (Figure 1a) [13], showing both strong expressions in EV fractions, and by Dot Blot against CD9 (Figure 1b). Transmission electron microscopy confirmed the presence of EVs in the BBT-9 and -18 secretome (Figure 1c), with a similar diameter (133, 150 nm), perimeter (466, 524 nm), and roundness (0.8) (Figure 1d).

We then analyzed, by mass spectrometry, the secretome of these early implantation stage embryonic BBTs, with special attention to changes in the secretion profile of proteins related to embryonic implantation at early pregnancy, angiogenic proteins, as well as changes resulting from an EMT.

We performed a comparative protein analysis between the EV fractions from BBT-9 and BBT-18 conditioned media and the same fractions from the non-conditioned culture medium (control). Although in all cases, FBS in culture media had been depleted from EVs by long-term ultracentrifugation, we could still detect in the proteomic analysis some remnant EV components in non-conditioned media. Since our cells are of bovine origin, these proteins cannot be filtered from the data. Yet, a total of 1231 proteins, out of 1577, were specific from BBT-conditioned media, and only 49 proteins were exclusively detected in the non-conditioned medium (Figure 2a). A total of 1090 protein groups were detected in EV samples, from which 711 were common and 184 and 195 proteins were differentially expressed in BBT-9 and BBT-18, respectively (Figure 2a).

The mass spectrometry analysis performed on the soluble protein fractions from SEC detected 1285 protein groups, from which 547 were common to both BBT samples, and 454 and 284 were differentially expressed in BBT-9 and BBT-18, respectively (Figure 2b), considering differences bigger than 2-fold in all comparative analyses.

These proteomic data were subjected to a bioinformatic analysis using the UniProt database. GO (Gene Ontology) analyses were performed using the PANTHER database (Appendix A). Classification of the identified proteins either in the soluble fraction or EV-cargo of BBT-9 and BBT-18, following biological processes, molecular function, and protein class, was performed, and differences were observed between BBT-9 and BBT-18 in both the soluble fraction and EV-cargo proteins in protein families related to structural molecule activity, biological adhesion, or reproduction (Appendix A).

Interestingly, more detailed scrutiny of the BBT-9 secretome revealed the presence of the EMT markers, transforming growth factor beta (TGFB), Collagen alpha-2(I) chain (COL1A2), and Vimentin (VIM) in both the soluble and the EV fraction, while fibroblast growth factor receptor (FGFR1) and 72 kDa type IV collagenase (MMP2) were exclusively present in the soluble fraction (Figure 2c). The epithelial marker Cadherin-1 (CDH1) and the pregnancy epithelial marker mucin-1 (MUC1) were exclusively expressed in the BBT-18 soluble fraction and EVs, respectively (Figure 2c and Table 1).

### 2.2. The BBT-Secretomes Contain Early Pregnancy-Related Proteins and Angiogenic Proteins Both in the Soluble Fraction and EV-Cargo

The proteomic analysis also identified 62 early pregnancy-related proteins (EPRPs), 41 in EV fractions (Table 1 and Table 2), and 21 soluble proteins (Table 3). BBT-18 and BBT-9 overrepresented 13 and 8 EPRP in the soluble fraction (Figure 3a) and 12 and 7 EPRP in EVs (Figure 4b), respectively. From the set of 41 EPRP proteins in EVs, we identified a group of 18 proteins corresponding to a day 16 conceptus, commonly expressed in BBT-18 and BBT-9 (Table 2).

Interestingly, we could also detect a series of proteins related to angiogenic pathways (Table 4), five of them found in both BBT-9 and BBT-18 secretomes, four overrepresented in BBT-9, and two overrepresented in BBT-18.

### 2.3. Endometrial and Blood MSCs Show Chemotactic Migration to the Secretome of the Embryonic Trophectoderm Cells

We aimed to build an in vitro model of cell-to-cell communication through EVs and soluble proteins. To demonstrate the functional transfer between embryonic trophectoderm and maternal MSC of protein-containing secretome, we first confirmed that EVs isolated from conditioned media of BBT cells previously labeled with PKH26 were successfully uptaken by maternal eMSC and pbMSC during a chemotaxis experiment in an agarose spot assay following the procedures of Calle et al. [11,15] and Monguió-Tortajada et al. [16] as described in materials and methods (Appendix A).

To test the functionality of this maternal–embryonic communication system, we analyzed the chemotactic effect of BBT secretome on the migration of maternal eMSC and pbMSC using an agarose spot assay. All MSC lines increased their migration towards BBT-9- or BBT-18-soluble proteins or EVs in terms of maximum Euclidean distance (MED) except pbMSC towards BBT-18-EVs (Figure 4a,b, and Appendix A). It should be taken into account that without strong chemotactic stimuli, the physical constraints imposed by the agarose spot impede the migration of MSCs towards the center of the drop so that in a rose diagram (Figure 4a), it appears as if MSCs were repelled by the drop and migrated to the opposite direction so that the Center Of Mass of the population appears ≥ 74 µm outside the edge of the drop (Figure 4c,d). In contrast, when soluble proteins or EVs were used as stimuli, MSC lines showed their COM near the edge or inside the agarose drop (<65 µm). pbMSCs were the most migratory lines when they were stimulated with soluble protein from both BBT-9 and BBT-18, while they were also the fastest without stimulation or with protein stimulation from BBT-9, EV-cargo, or soluble protein (Figure 4e).

### 2.4. Embryonic Trophectoderm EVs Change Their Implantation Protein Profile after Uptake of Maternal MSC-EVs

Since maternal/embryo communication is a bidirectional phenomenon, we next analyzed the changes in the secretome of BBT as a result of the uptake of either eMSC- or pbMSC-EVs or after treatment with the endometrial cytokines Activin A and follistatin (FLST1). Activin A and FLST1 are present in the uterine fluid during the peri-implantation period. Activin A induces EMT in human and bovine trophoblast cells and FLST1 is an inhibitor of Activin A. To analyze the interaction between activin A and FLST1 in bovine trophoblasts with epithelial or mesenchymal phenotype, BBT-9 or BBT-18 cells were treated with Activin A or Activin A + FLST1. MSC-EVs were isolated by SEC from conditioned media of eMSC or pbMSC stimulated or not during 48 h with IFN-τ. Then, EVs from the different lines and conditions were pooled separately. BBT-9 and BBT-18 were then cultured for 48 h in media supplemented with the different EVs or in the presence of Activin A, Activin A, and FSTL1 or without cytokines (Figure 5). EVs were then isolated from the conditioned media of BBT cultures. The amount of EVs secreted by trophectoderm cells was analyzed by quantifying the CD9 signal in a dot blot using ImageJ software (Figure 6a). Activin A or BBT-18-EVs treatments showed CD9 overexpression ≥ 2-fold.

We then focused on a series of proteins related to the process of invasive growth and involved in the regulation of embryo implantation and analyzed them on BBT-derived EVs by bead-assisted flow cytometry. We observed constitutive secretion in both BBT-9- and BBT-18-EVs of PEG3 implantation protein as well as high recovery of HSPH1 in BBT-18-EVs (Figure 6b,c). When the BBTs’ primary cultures were stimulated with Activin A or with the combination of Activin A and FLST1, their EV cargo now contained the full repertoire of implantation proteins assessed: MMP2, TDGF1, HSPH1, and PEG3.

Regarding stimulation with EVs from MSC, bigger changes were observed in BBT-9 when EVs were derived from eMSC than from pbMSC. Thus, eMSC-EV stimulation induced the incorporation of MMP2 and TDGF1 in BBT-EV cargo. In contrast, these changes were smaller when the trophectoderm cells were stimulated with EVs derived from eMSC that had been previously stimulated with IFN-τ. HSPH1 recovery in trophectoderm EVs showed very different behavior in BBT-9 and BBT-18 primary cultures. BBT-18 showed an exhibiting remarkable HSPH1 expression, which was further increased by Activin A stimulation, whereas all other stimuli impaired its accumulation in EVs, including the combination of Activin A and follistatin and all the MSC-derived EVs treatments.

## 3. Discussion

Recently, our group characterized for the first time endometrial mesenchymal stem cell lines (eMSC) [11] as well as peripheral blood MSC (pbMSCs) lines [12]. While eMSC showed a reduced migratory capacity in response to the implantation cytokine IFN-τ [11], pbMSC showed chemotactic behavior to both inflammation (TNFα, IL1β), embryo implantation stimuli (IFN-τ) or BBT-secretome, suggesting that the embryo secretome plays a role in ensuring the retention of eMSC and the active recruitment of MSCs from bone marrow during early pregnancy to repress the immune response to prevent the embryo rejection by the maternal organism [12].

During bovine pregnancy, trophoblast adhesion and placental formation have been reported to require a gradual loss of epithelial characteristics but without the acquisition of full mesenchymal characteristics. [1]. This partial EMT concurs with gene expression changes associated with EMT in the bovine trophectoderm following conceptus attachment to the luminal epithelium [1]. After conceptus implantation, the trophectoderm looses the adherents’ junction molecule, CDH1, and gains the expression of mesenchymal markers, such as VIM and CDH2, maintaining, however, the expression of the epithelial marker cytokeratin. Trophectoderm EMT was thereafter shown to be regulated by the endometrium via activin A and FLST1 on days 20–22 [5]. Trophoblast binucleate cells in the bovine placentome showed a cytoplasmic distribution of CH1 and β-catenin translocation into the nucleus [17], suggesting a role for CDH1–β-catenin axis in trophoblast differentiation. On day 22, trophoblast CDH2 is highly expressed so that an increment in CDH1 degradation could also be involved in the further reduction to its expression. Hence, the loss of CDH1 as the conceptus attaches to the luminal epithelium may play a crucial role in the gene expression transition required for the successful progression from implantation to placentation. Regarding our BBT primary cultures, we observed that the expression of CDH1 was overrepresented in BBT-18 soluble fraction but not in BBT-9, while BBT-9-secretome is expressing VIM while maintaining cytokeratin expression. These parameters would reflect the two abovementioned phenotypes: epithelial and mesenchymal, respectively. CDH2 was expressed in vitro when trophectoderm cells were co-cultured with endometrial epithelial cells [18] but has not been reported in monocultures of trophectoderm cell lines, possibly explaining the lack of detection of CDH2 in the BBT-secretome.

EMT induction is regulated at the molecular level by a variety of growth factor signals—in particular, transforming growth factor β (*TGF-β*), hepatocyte growth factor (*HGF*), epidermal growth factor (*EGF*), fibroblast growth factor (*FGF*), Wnt proteins, and IL-6 [19]. *TGF-β* is a multifunctional cytokine that is considered the main inducer of EMT. The *TGF-β* signaling pathway plays an important role in regulating cell proliferation, differentiation, invasion, migration, apoptosis, and microenvironmental modification, and stimulates pathophysiological EMT and metastasis [20]. *TGF-β* and *FGFR1* proteins were found to be overrepresented in BBT-9-secretome, suggesting that these proteins could be responsible for the regulation of EMT in the bovine trophectoderm. In addition, it has been reported that *TGF-β* is a soluble factor produced by MSCs, which mediates the suppression of T-cell proliferation [21]. Therefore, the BBT-9, thanks to its mesenchymal phenotype, could play an immunomodulatory role via EV signaling, aimed at guaranteeing the survival of the embryo. Our group has also reported the immunomodulatory capacity of eMSC and pbMSC by inhibiting the proliferation of human T lymphocytes [11,12].

During EMT, *FGF* increases the expression of vimentin and induces the activity of *MMP2*, increasing cell mobility. *FGF* also causes changes in the actin cytoskeleton allowing anchorage-independent growth [22]. Vimentin and *MMP2* are overrepresented in BBT-9-EVs and BBT-9-soluble fractions, respectively. Although bovine trophoblasts do not penetrate the endometrium, upregulation of *MMP2* metalloproteinase suggests that it could play a role in the non-invasive trophectoderm. *FGF1*, the ligand for *FGFR1*, overrepresented in BBT-9-soluble fraction, is also known to upregulate *MMP13*, resulting in EMT induction [23].

Other characteristic EMT markers, such as the transcription factors, *SNAI2*, *ZEB1*, *ZEB2*, *TWIST1*, *TWIST2*, and *KLF8S*, were not found in EVs or soluble proteins from BBT-secretome, as we expected since they are nuclear proteins. Likewise, we were also unable to detect pluripotency markers (*OCT4/POU5F1*, *SOX*, *NANOG*) or characteristic markers of trophectoderm (*ELF5*, *EOMES*, *GATA5*) in EVs or soluble proteins from BBT-cargo, although some of the mentioned markers could be detected by RT-PCR in our BBTs [10].

After EMT, a micro-angiogenesis process related to uterine vascularization is necessary for adequate implantation [24] and placenta formation. We were able to detect eleven proteins involved in angiogenesis pathways in the BBT secretome, from which six participate in the vascular endothelial growth factor (*VEGF*) pathway. The *VEGF* signaling pathway is essential for all stages and processes involved within the vascular development (vasculogenesis, angiogenesis, and lymphangiogenesis). *VEGF* is the main factor that regulates angiogenesis in bovine pregnancy. Interestingly, the BBT-9 secretome showed an overrepresentation of several angiogenic-related factors: fibroblast growth factor receptor 1 (*FGFR1*), RHO GTPase-activating protein 1 (*ARHGAP1*), RHO-related GTP-binding protein (*RHOC*), and vascular cell-adhesion molecule (*VCAM1*). However, in BBT-18 secretome, only the Serine/threonine-protein kinase A-RAF (*ARAF*) and Heat Shock Protein Beta-1(*HSPB1*) from this pathway were found to be overrepresented.

In sum, both the overexpression of EMT markers and the higher detection of angiogenic factors in the BBT-9 protein profile would support the notion that BBT-9 is exhibiting a more advanced developmental stage near to an EMT stage. However, we could not detect in the secretome of BBT the typical bovine pregnancy angiogenic related markers such as *VEGF* family proteins or its receptor, Angiopoietin (*ANGPT*)-2/*ANGPT-1* [24,25] due likely to an early developmental stage.

In the bovine species, the expression of integrins (ITGs) has been characterized at the uteroplacental interface during the periods of trophectoderm attachment and placentation [26]. *ITG**αV* (overrepresented in BBT-9-EVs cargo), in combination with the *β5* subunits, is known to bind to Osteopontin (*SPP1*) [27]. *ITG**β1* is also overrepresented in BBT-9-EVs cargo. It can form heterodimers with the *α4* chain, also known as very late antigen-4 (*VLA4*) often detected in mesenchymal stem cells [28], and *α8* subunit resulting in alternative receptors for osteopontin *SPP1* [29]. Yamakosi et al. observed that the subunits of *SPP1*-binding ITGs are upregulated during the embryo attachment process and indicate their possible involvement in the trophoblast adhesion to the endometrial epithelium in cows [1]. Moreover, ITG on EVs was reported to dictate organ-specific uptake of EVs to initiate pre-metastatic niche formation in a tumor scenario [1]. Analogously, integrin expression profiles of trophectoderm-secreted exosomes could be relevant to direct maternal MSCs to the implantation niche and could be used as prognostic factors to predict good maternal immunoregulation to avoid embryo rejection.

As we have described previously, a correct balance that coordinates active immunity and tolerance during the contact between mother and conceptus is critical. Moreover, it is known that MSCs from the placenta or decidua are involved in the induction of this maternal immune tolerance [30,31]. In mice, MSCs from bone marrow are involved in the reduction of the embryo resorption rate by regulating the function and phenotype of macrophages and T cells at the maternal–fetal interface [32]. *ITG*
*α4β1* major counter ligand, vascular cell adhesion molecule-1 (*VCAM-1*), plays an important role in leukocyte recruitment during an immune response [33]. Bai et al. reported that uterine *VCAM-1* expression was minimal in day 17 cyclic and pregnant animals but increased between days 20 and 22 of pregnancy [28]. The authors also reported that *VCAM-1* expression in CT-1 cells (a trophoblast primary culture seeded onto STO mouse feeder cells [1]) was up-regulated with the use of uterine flushings. *VCAM-1* is found to be overrepresented in BBT-9-EVs cargo. Galectin 3 (*LGALS3*) also plays central roles in immune system regulation, shaping both innate and adaptive responses both in physiological and pathological processes [34]. Galectin 3, which is overrepresented in BBT-9-EV cargo, is first expressed in the trophectoderm cells of the implanting embryo and has been implicated in the process of implantation [35].

We have demonstrated that both eMSC and pbMSCs are capable of migrating in response to BBT chemotactic stimuli reinforcing the idea of their participation in pregnancy establishment. In our experiments, we employed eMSC from heifer donors. In addition, pbMSCs were derived from male blood samples. Although these materials were easier to obtain, previous studies have shown that the gender or age of MSC does not affect their secretion of trophic factors [36]. In addition, working with cells derived from young specimens also ensures that they have never been exposed to a pregnancy environment. Trophoblastic cells were derived from hatched embryos since later stages of development are not feasible in pure in vitro systems. Although the different BBT cell lines obtained present different characteristics, some of them resembling those occurring in later stages of embryo development, it would be interesting to analyze the chemotactic stimuli secreted by trophoblastic cells derived from elongated embryos. Considering that BBT-9 shows developmental features nearer to EMT than BBT-18, it would suggest that it represents a more advanced developmental stage, closer to embryo implantation than BBT-18. Therefore, our chemotactic data would support that epithelial embryonic trophectoderm (BBT-18) stimulates chemotactic migration of maternal MSCs from the endometrium through soluble and EVs mediators, while it attracts peripheral MSC only through soluble mediators. In contrast, when the embryonic trophectoderm already presents mesenchymal characteristics (BBT-9), it stimulates the migration of endometrial or stimulates peripheral maternal MSCs through both soluble and EV-cargo proteins, traveling long distances and at high speed. Therefore, at late implantation stages (BBT-9) secretome-dependent signaling could provoke a highly intensified call effect in MSCs to ensure embryo implantation at that critical point of pregnancy.

We next analyzed the changes in the secretome of embryonic trophectoderm as a result of the uptake of maternal MSC-EVs or treatment with endometrial cytokines present in the uterine fluid during the peri-implantation period. Activin A is a known member of the *TGF-ß* superfamily, and *FSLT1* is an inhibitor of activin A [37]. Kusama et al. reported that *FLST1* increased on day 20 uterine flushing and decreased on day 22, whereas they found an elevated activin A expression on day 20 and a further increase on day 22. [5]. Kusama et al. reported Activin A-induced EMT markers expression was inhibited by *FLST* in the embryonic trophectoderm cells CT-1 [5]. To analyze the interaction between activin A and *FLST1* in bovine trophoblasts with epithelial or mesenchymal phenotype, BBT-9 or BBT-18 cells were treated with Activin A or Activin A + *FLST1*.

As a readout, we decided to focus on the possible variations whithin the secretion pattern of a selected group of proteins related to the process of invasive growth and thus potentially involved in the regulation of embryo implantation: *TDGF1*, *HSPH1*, *MMP2*, and *PEG3* [38,39,40,41]. Our embryonic trophectoderm primary cultures either with epithelial or mesenchymal phenotype responded to Activin A or Activin + *FLST1* stimulations by secreting the implantation proteins in EV-cargo. Mitko et al. reported a marked increase in the expression of *TDGF1* at day 12, connected with the process of invasive growth and regulating the embryo implantation [39]. Hatayama et al. reported a marked increase in the expression of *HSPH1* in mouse embryos between days 9 and 12, coinciding with organogenesis, and they attributed to *HSPH1* a relevant function in organogenesis during embryonic development [42]. Later, Yuan et al. also reported the presence of *HSPH1* in rat embryos [43]. In our in vitro model of communication between maternal MSCs and embryonic trophectoderm through EVs, we observed a marked expression of *HSPH1* in EVs from trophectoderm with the epithelial phenotype (BBT-18), which was further increased in the presence of Activin A. In contrast, the expression of *HSPH1* in EVs from embryonic trophectoderm with mesenchymal phenotype was much lower. MMP2, a member of the matrix metallopeptidase family, which is involved in the breakdown of extracellular matrix in normal physiological processes, and *PEG3* are up-regulated on Day 13 in vivo conceptuses [1].

## 4. Materials and Methods

### 4.1. Cells

Bovine blastocyst embryonic trophectoderm primary cultures (BBT): BBT-9 and BBT-18 primary cultures were established in our laboratory from an embryo biopsy of good quality bovine hatched blastocysts produced in vitro [10].

Endometrial mesenchymal stem cell (eMSC) lines: eMSC-1A, -3A, -3D, -3E, -4B, -4C, -4D and -4H were isolated, established, and immortalized (PA-317 LXSN-16E6E7 cells) in our laboratory from the uterus of heifers at different oestrous cycle stages [11].

Peripheral blood mesenchymal stem cell (pbMSC) lines were isolated and established in our laboratory from male heparinized whole blood. The pbMSC-80, -81, and -84 cell lines were obtained by previous bone marrow mobilization using granulocyte colony-stimulating factor (G-CSF), and pbMSC-922, -923 without it [12].

### 4.2. Production and Isolation of BBT Primary Cultures (BBT-9 and BBT-18) Secretome (EVs and Soluble Proteins)

The proceedings for the isolation and quantification of the secretome produced by BBT-9 and BBT-18 primary cultures were performed as detailed in [12]. FBS was depleted of bovine EVs by ultracentrifugation at 100,000× *g* for 16 h (Sorvall AH- 627 rotor, L8–70M ultracentrifuge, Beckman). Conditioned media from BBT cultured with 10% EVs depleted FBS, was collected after 72 h of confluent culture, centrifuged at 2.000× *g* for 30 min, and concentrated by centrifugation at 2.000× *g* for 50 min using Amicon Ultra-15 Centrifugal Filter Units (Millipore, Billerica MA). EV production and isolation were carried out following the procedures of Suarez et al. with minor modifications [13]. Briefly, the whole supernatant (±150 μL) was loaded onto a 1 mL Sepharose CL-2B (Sigma Aldrich, St. Louis, MO, USA) size exclusion chromatography (SEC) column. Elution was performed by gravity adding PBS, collecting 20 sequential fractions of 100 μL. The presence of EVs in the collected fractions was detected by dot blot and bead-assisted flow cytometry using typical exosome markers (CD9 and CD63); EVs started to be eluted in fraction 3.

### 4.3. Characterization of BBT Primary Cultures (BBT-9 and BBT-18) EVs by Bead-Assisted Flow Cytometry Assay

The selection of SEC fractions enriched in EVs was performed by bead-based flow cytometry analysis of each fraction using anti-CD63 (CC25 Bio-Rad, Barcelona, Spain) and anti-CD9 (VJ1/20) antibodies (Figure 1). The three fractions with the highest MFI values for these EV markers (commonly 3rd–6th) were pooled for further EV downstream analyses. Next, 10 μL of each fraction isolated by SEC were incubated with 0.25 μL of aldehyde/sulfate latex beads (ø = 4 μm; 5.5 × 10^6^ particles/mL; Invitrogen, Carlsbad, CA, USA) for 15 min at RT. Then, 1 mL of PBS supplemented with 0.1% BSA ((PBS-BSA); Thermo Fisher Scientific) was added, and the sample was incubated overnight on rotation. Bead-coupled EVs were pelleted by centrifugation at 2000× *g* for 10 min, washed with 1 mL of PBS-BSA, and centrifuged again. The pellet was resuspended with 50 μL of PBS-BSA per analysis and stained using murine hybridoma supernatant of anti-CD9 (VJ1/20) and anti-Bovine-CD63 (CC25 Bio-Rad) as primary antibodies and FITC-conjugated secondary antibodies (Thermo Fisher Scientific) [44]. Negative control was obtained by incubating the beads coupled with the EVs sample, in the absence of primary Ab. Washing steps were performed once after primary and twice after secondary Ab labeling with 1 mL of PBS-BSA and centrifugation at 2000× *g* for 10 min. Data were acquired in a conventional flow cytometer (FACSCanto A, BD Biosciences, San Jose, CA, USA) and analyzed with the FlowJo software (version Tree Star, Ashland, OR). Gating of EV-decorated 4 μm diameter beads was performed based on FCS/SSC parameters so that unbound EVs or possible antibody aggregates are excluded from the analysis.

The selection of BBT-9 and BBT-18 SEC fractions enriched in soluble proteins was performed by quantification with a Nanodrop spectrophotometer (Thermo Fisher Scientific, Pittsburgh, PA, USA) at 280 nm absorbance using 2 μL per fraction. The SEC fractions with the highest soluble protein content were selected and pooled.

### 4.4. BCA Protein Analysis

The protein concentration of EVs and soluble protein SEC fractions pooled was assessed following the Pierce^®^ BCA Protein assay kit protocol (Thermo Fisher Scientific, Pittsburgh, PA, USA). Duplicates of the sample’s absorbance at 540 nm were measured in a Tecan GENios Microplate reader.

### 4.5. Dot Blot Analysis

EV samples were directly dispersed onto a nitrocellulose membrane (GE LifeSciences, Amersham, Germany). Membranes were blocked with 10% skimmed milk and incubated with anti-CD9 (VJ1/20) [44] followed by peroxidase-coupled secondary antibodies and detected by chemiluminescence with an ImageQuant LAS500 biomolecular imager (GE LifeSciences, Amersham, Germany).

### 4.6. Transmission Electron Microscopy (TEM)

To assess the size and morphology of EVs using transmission electron microscopy, ionized carbon and collodion-coated copper electron microscopy grids were floated on a sample drop, washed, and stained with 2% uranyl acetate (in double-distilled water) for 1 min, and visualized in a JEM-1010 (JEOL, Tokyo, Japan) transmission electron microscope. A quantitative evaluation of the BBBT-18 and BBBT-9 trophectoderm primary cultures in terms of particle diameters was determined by the computer-assisted software TEM ExosomeAnalyzer [45]. 

### 4.7. BBT-EV Binding to MSC

To assess the BBT-EV and MSC interaction, a 2D chemotaxis assay was developed using PKH26-labeled BBT-9 and BBT-18 according to the manufacturer’s instructions. Briefly, the labeling of cells with PKH26 (Red Fluorescent Cell Linker Kits MINI26; Sigma-Aldrich Co., St Louis, MO, USA) was performed for 5 min at RT in the dark and blocked with FBS. The unincorporated stains were removed by BBT centrifugation at 400× *g* for 10 min at RT using Heraeus Biofuge Primo R Centrifuge. The BBTs were washed with PBS and subjected to additional centrifugation. The pellet was resuspended, and labeled cells were plated in a 100 mm^2^ tissue culture dish (JetBiofil, Guangzhou, China) and incubated in an atmosphere of humidified air and 5% CO_2_ at 37 °C for 72 h. Culture media were supplemented with 10% of EV-depleted FBS. EVs were then isolated by SEC from BBT conditioned media. A cell migration analysis by agarose spot assay with labeled EVs from BBT was carried out following the procedures of the 2D chemotaxis assay described below in materials and methods. A control with BBT-EVs from unlabeled BBT was performed. After O/N incubation, MSCs were visualized in fluorescence microscopy.

### 4.8. Identification and Quantification of Secretome (EVs and Soluble Fraction- Derived Proteins of BBT Cells by LC-MS/MS

To achieve the most accurate characterization of soluble protein fractions for the HPLC technique, we did not add FBS to BBT-9 and BBT-18 culture media to produce conditioned media without FBS. This allowed us to avoid possible confusion between bovine proteins contained naturally in the FBS and proteins secreted by BBT cells. In the case of EV protein samples and the control group from culture media, FBS was EV depleted and analyzed by HPLC.

### 4.9. Sample Preparation for LC-MS/MS

After denaturation of protein with 8 M urea in 50 mM ammonium bicarbonate pH 8.8, samples were reduced and alkylated; briefly: disulfide bonds from cysteinyl residues were reduced with 10 mM DTT for 1 h at 37 °C, and then thiol groups were alkylated with 10 mM iodoacetamide for 30 min at room temperature in darkness. Samples were diluted to reduce urea concentration below 1.4 M and digested using sequencing grade trypsin (Promega, Madison, WI, USA) overnight at 37 °C using a 1:20 (*w*/*w*) enzyme:protein ratio. Digestion was stopped by the addition of 1% TFA. EV samples whole supernatants were dried down and then desalted onto Pierce Peptide Desalting Spin Columns (Thermo), and secretome samples in OASIS C18 columns (Waters), until the mass spectrometric analysis. Protein Identification by reverse phase-liquid chromatography-mass spectrometry analysis (RP-LC-MS/MS), Data processing and Protein quantification by iTRAQ-LC-MS/MS labeling and high pH fractionation, detailed in Appendix A.

### 4.10. In Vitro Model of Chemotactic Migration of MSC by BBT Secretome

The cell migration measurement by agarose spot assay was carried out following the procedures of Calle et al. [11,15]. A 0.5% agarose solution in PBS was heated on a water bath until boiling to facilitate complete dissolution. When the temperature cooled down to 40 °C, 90 μL of the agarose solution was pipetted into a 1.5 mL Eppendorf tube containing 10 μL of PBS, EVs (0.2 µg/µL), or soluble proteins (0.2 µg/µL)) from BBT-9- or –BBT-18. Five-microliter spots of agarose-containing PBS, EVs or soluble proteins were pipetted onto 12-well plates (JetBiofil, Guangzhou, China) and allowed to cool down for 15 min at 4 °C. At this point, 2.5 × 10^5^ cells were plated carefully onto spot-containing wells in the presence of culture media. After a period of resting within the cells attached to the plate in the incubator, imaging was performed in temperature, humidity, and CO_2_ controlled chamber for 20 h by FRET microscope (Zeiss Axiovert200) coupled to a monochrome digital camera (Hamamatsu C9100-02). Pictures were taken each 10 min and processed using Metamorph 7.10.1.16 software. We measured different parameters to achieve a deep description of cell migration, such as Maximum Euclidean distance (MED) to the stimuli. This parameter represents the maximum length of the straight traveled distance by cells (µm) from the edge of the spot to the center, which is the reference point for the stimuli position. Moreover, we also tracked the random choice of cells present at the edge of the drop at the beginning of the experiment to measure the mean speed (µm/min), the accumulated distance, the center of mass (COM) that is the average position of starting and end cell points of all tracked cells, and, finally, the density of cell endpoints represented by Rose diagrams showing the most common direction taken for motile cells. The MED from the border of the spot was measured for each drop using Image J free software. The rest of the evaluated parameters were measured using the Chemotaxis and Migration Tool (free software IBIDI) combined with the manual tracking plugin of Image J. We have always tracked the cells in the same drop quadrant to avoid differences in the reference point. All experiments were replicated at least 3 times.

### 4.11. Analysis of Changes in BBT EVs Profile after Uterine Cytokines Stimuli or Communication with MSC EVs

To assess the changes in implantation protein profiles in BBT-EVs as a result of BBT-MSC in vitro communication, pre-confluent 100 mm^2^ tissue culture dishes of BBT-9 and BBT-18 were cultured for 48 h in media supplemented with MSC-EVs isolated by SEC from stimulated or unstimulated MSC. Confluent 100 mm^2^ tissue culture dishes of eMSC or pbMSC were cultured for 48 h with or without IFN-τ stimulation (MBS1131960, Mybiosource) (1000 ng/mL). Then, EVs from the different eMSC or pbMSC lines were pooled separately. In addition, to assess uterine cytokines effect on BBT-EVs, pre-confluent 100 mm^2^ tissue culture dishes of BBT-9 and BBT-18 were cultured in media supplemented with Activin A (PHC9564, Thermo Fisher Scientific) (100 ng/mL), Activin A, and *FSTL1* (10924H08H50, Thermo Fisher Scientific) (100 and 50 ng/mL, respectively) or without cytokines during 48 h. The experimental design is shown in Figure 5. Bead-assisted flow cytometry was performed on the BBT-EVs obtained following the procedures described to analyze the implantation marker expression of *MMP2* (MA1-772, Thermo Fisher Scientific); *HSPH1* (PA5-77793, Thermo Fisher Scientific); *TDGF1* (NB100-1597, Novus Biologicals), and *PEG3* (TA343590, Acris-antibodies).

### 4.12. Statistical Analysis

All experiments were repeated with at least three independent biological replicates. One-way ANOVA with Tukey multiple comparison tests and two-way ANOVA with Sidak multiple comparison tests were performed. All results are expressed as mean ± SEM or as median (range). All the analyses and relative graphs were made in Prism 9.0 (GraphPad Software Inc., La Jolla, CA, USA). For the statistical analysis of rose diagrams, we have used the Rayleigh test. This is a statistical test for the uniformity of a circular distribution of cell endpoints included in Chemotaxis and migration tool software.

## 5. Conclusions

We have confirmed different epithelial–mesenchymal transition stages in embryonic trophectoderm primary cultures. The embryonic trophectoderm secretome contains early pregnancy-related proteins and angiogenic markers both as cargo in EVs and the soluble fraction. We have demonstrated the functional transfer of protein-containing secretome between embryonic trophectoderm and maternal MSC and its chemotaxis capacity, thus suggesting that this system could be used as an in vitro model of cell-to-cell communication through EVs and soluble proteins. In a complementary way, the pattern of secretion of implantation proteins in trophectoderm-EV changes depending on: (i) its epithelial or mesenchymal phenotype; (ii) as a result of the uptake of eMSC- or pbMSC-EVs previously stimulated or not with embryonic signals; (iii) because of stimulation with endometrial cytokines present in the uterine fluid in the peri-implantation period.

Figure 7 shows a chronologic schematic representation of the BBT-identified proteins associated with the main reproductive or EMT events.

## Figures and Tables

**Figure 1 ijms-22-05638-f001:**
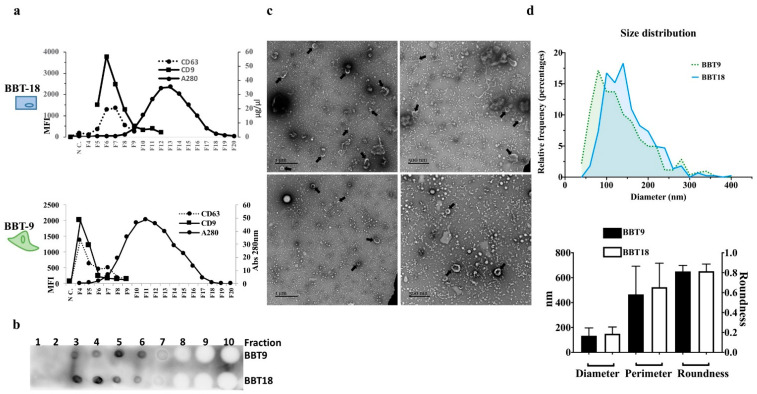
Bovine blastocyst embryonic trophectoderm primary cultures release EV-cargo proteins in homogeneous populations of EVs and soluble proteins. Representative size exclusion chromatography (SEC)-elution profile of EVs from BBT-9 and BBT-18 analyzed by bead-assisted flow cytometry using anti-CD63 and anti-CD9 antibodies. Mean fluorescence intensity (MFI) relative to the negative control is plotted in the left *y*-axis (**a**). Protein concentration was analyzed by Nanodrop for each fraction and plotted on the right *y*-axis. (**a**). SEC-elution profile of EVs from BBT-9 and BBT-18 analyzed by Dot Blot using an anti-CD9 specific antibody (**b**). Fresh BBT-9-EVs or BBT-18-EVs isolated were negatively stained with uranyl acetate and visualized by TEM (**c**). Quantitative results including diameter, perimeter, and roundness of BBT-9-EVs and BBT-18-EVs (**d**).

**Figure 2 ijms-22-05638-f002:**
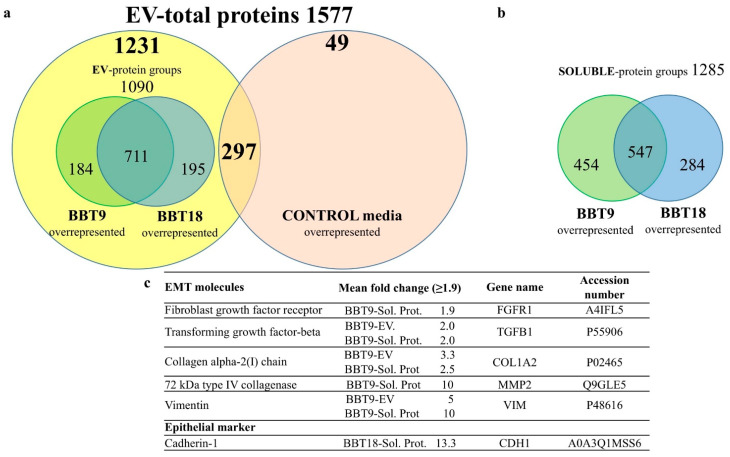
BBT-9- and BBT-18-derived secretomes contain EV-cargo proteins and soluble proteins. Venn diagrams of overrepresented EV-cargo proteins (**a**), or soluble proteins (**b**) identified by iTRAQ-LC-MS/MS analysis, overexpression ≥ 1.9-fold has been considered in all comparative analyses. Characteristic EMT expression markers identified and overrepresented in BBT-9-secretome and epithelial marker CDH1 overrepresented in BBT-18-soluble fraction (**c**).

**Figure 3 ijms-22-05638-f003:**
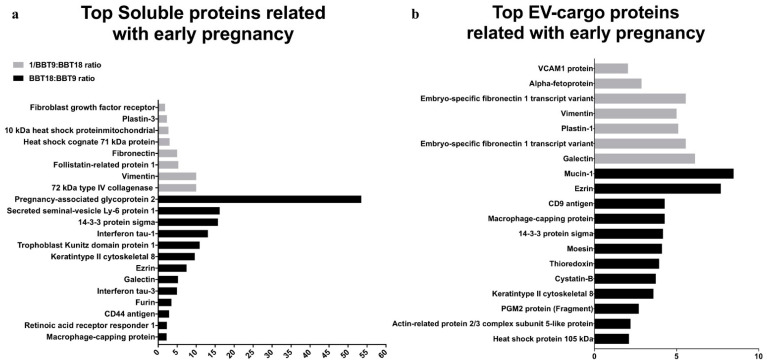
Bovine blastocyst embryonic trophectoderm primary cultures release soluble proteins (**a**) and EV-cargo proteins (**b**) associated with early pregnancy.

**Figure 4 ijms-22-05638-f004:**
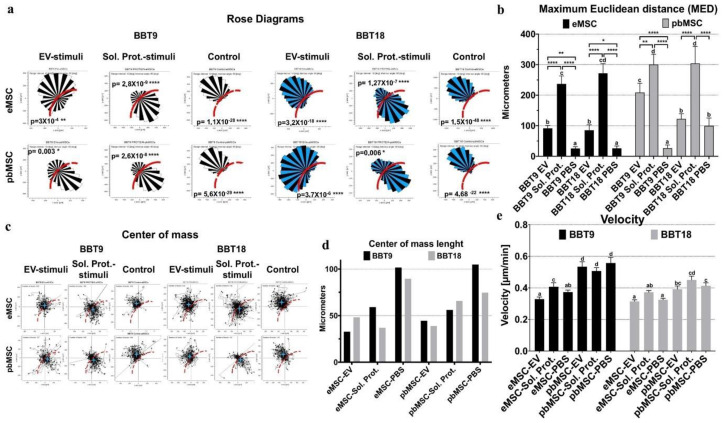
Maternal MSCs show chemotactic migration to BBT-9- or BBT-18-secretome. Analysis of the chemotactic effect on rose diagrams of eMSC and pbMSC without stimuli (control) or stimulated by EVs or soluble proteins from BBT-9- or BBT-18-secretome (**a**). The maximum Euclidean distance (MED) of eMSC or pbMSC towards EVs or soluble proteins from BBT-9 or BBT-18 was measured (mean ± SD). Different letters indicate a significant difference. * *p* < 0.05; ** *p* < 0.005; **** *p* < 0.0001. (**b**). The center of mass (COM): a strong chemotactic migration parameter for the analysis of the migrating response of eMSC and pbMSC towards EVs or soluble proteins from BBT-9 or BBT-18. Charts showing the trajectory plots marking each cell track with its endpoint (**c**), or the center of mass length (**d**), of eMSC and pbMSC without stimuli (control) or stimulated by EVs or soluble proteins from BBT-9 or BBT-18. The velocity of eMSC or pbMSC towards EVs or soluble proteins from BBT-9 or BBT-18 was measured (mean ± SD). Different letters indicate a significant difference (**e**).

**Figure 5 ijms-22-05638-f005:**
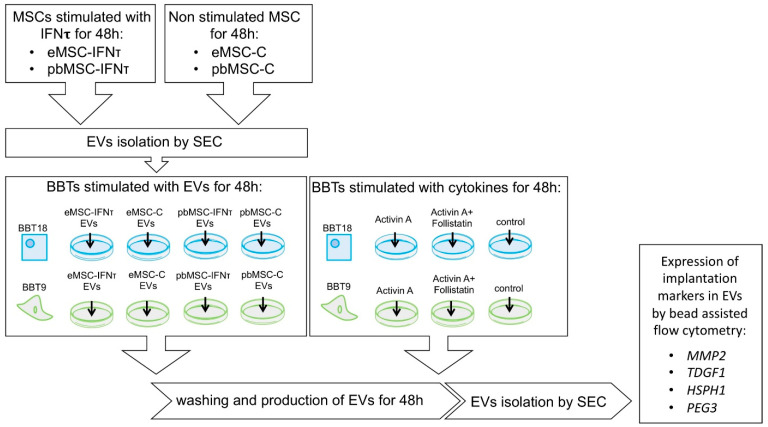
Experimental design.

**Figure 6 ijms-22-05638-f006:**
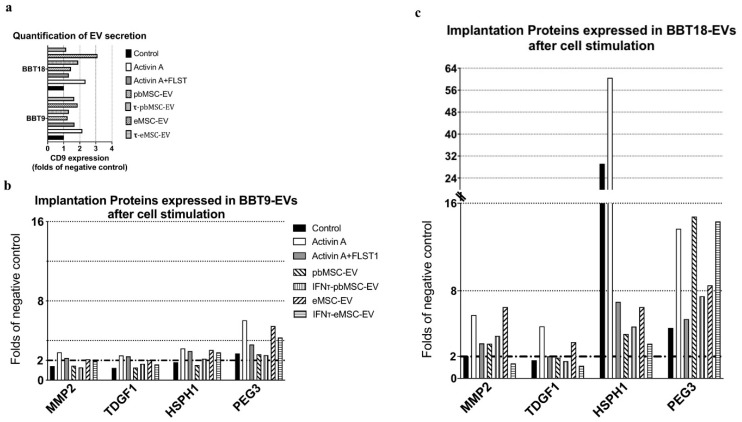
BBT secret implantation proteins on EVs, and their secretion pattern, which is altered after the uptake of MSC-EVs or the presence of uterine cytokines. SEC-elution profile of EVs from BBT-9 and BBT-18 for the different experimental conditions (Control, Activin A, Activin A+FLST, pbMSC-EVs, τ-pbMSC-EVs, eMSC-EVs, and τ-eMSC-EVs) were analyzed by Dot Blot using an anti-CD9 specific antibody. The positive EV fractions of each experimental condition were quantified by ImageJ software analysis of the dot blot results (**a**). Bead-assisted flow cytometry analysis of selected implantation proteins expressed in BBT-9- (**b**) and BBT-18-EVs (**c**) after cell stimulation.

**Figure 7 ijms-22-05638-f007:**
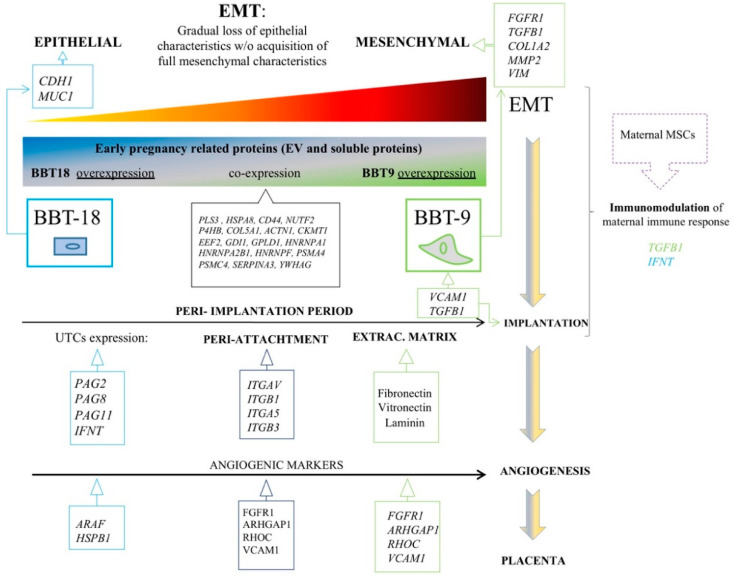
Representation of the BBT-identified proteins associated with the main reproductive or EMT events.

**Table 1 ijms-22-05638-t001:** Top early pregnancy-related proteins differentially abundant between BBT-18- and BBT-9-EV-cargo detected by iTRAQ-LC-MS/MS analysis.

EV-Cargo Proteins	Mean Fold Change BBT-18:BBT-9	Gene Name	Accession Number	Reproductive Event
Mucin-1	8.47	MUC1	Q8WML4	Epithelial. Implantation
Ezrin	7.69	EZR	P31976	Implantation
Macrophage-capping protein	4.27	CAPG	Q865V6	Co. secretion d16 pregnant cow.
CD9 antigen	4.27	CD9	P30932	Implantation (in vitro)
14-3-3 protein sigma	4.17	SFN	Q0VC36	Trophoblast marker
Moesin	4.11	MSN	Q2HJ49	Co. secretion d16 pregnant cow.
Thioredoxin	3.94	TXN	O97680	Co. secretion d16 pregnant cow.
Cystatin-B	3.73	CSTB	A0A140T831	Co. secretion d16 pregnant cow.
Keratin type II cytoskeletal 8	3.59	KRT8	P05786	Implantation
PGM2 protein (Fragment)	2.7	PGM2	A6QQ11	Co. secretion d16 pregnant cow.
Actin-related protein 2/3 complex subunit 5-like protein	2.19	ARPC5L	Q5E963	Co. secretion d16 pregnancy cow.
Heat shock protein 105 kDa	2.1	HSPH1	Q0IIM3	Co. secretion d16 pregnancy cow.
Galectin	−6.12	LGALS3	A6QLZ0	Co. marker peri-elongation
Embryo-specific fibronectin 1 transcript variant	−5.56	FN1	B8Y9S9	Co. secretion peri-gastrulation
Plastin-1	−5.10	PLS	A6H74	Co. secretion d16 pregnant cow.
Vimentin	−5.00	VIM	P48616	Co. secretion and EMT
Embryo-specific fibronectin 1 transcript variant	−5.56	FN1	B8Y9S9	Co. secretion peri-gastrulation
Alpha-fetoprotein	−2.86	AFP	Q3SZ57	Co. secretion peri-gastrulation
VCAM1 protein	−2.04	VCAM1	A7MBB0	Implantation

**Table 2 ijms-22-05638-t002:** Early pregnancy-related proteins coexpressed in BBT-18- and BBT-9-EV-cargo detected by iTRAQ-LC-MS/MS analysis. * Pregnancy-related proteins reported on in vivo day 16 conceptus in literature [14].

Secreted Proteins	Mean Fold Change BBT-18:BBT-9 (≤1.5)	Gene Name	Accession Number
Plastin-3	1.8	PLS3 *	A7E3Q8
Heat shock cognate 71 kDa protein	1.5	HSPA8 *	P19120
CD44 antigen	−1.36	CD44 *	Q29423
Nuclear transport factor	−1.28	NUTF2 *	Q32KP9
Protein disulfide-isomerase	1.15	P4HB *	P05307
Collagen type V alpha 1 chain	−1.29	COL5A1 *	G3MZI7
Alpha-actinin-1 OS=Bos taurus	1.87	ACTN1 *	Q3B7N2
Creatine kinase U-type mitochondrial	1.53	CKMT1 *	Q9TTK8
Elongation factor 2 OS=Bos taurus	1.63	EEF2 *	Q3SYU2
Rab GDP dissociation inhibitor alpha	−1.42	GDI1 *	P21856
Phosphatidylinositol-glycan-specific phospholipase	−1.58	GPLD1 *	P80109
Heterogeneous nuclear ribonucleoprotein A1	1.23	HNRNPA1 *	P09867|
Heterogeneous nuclear ribonucleoproteins A2/B1	−1.21	HNRNPA2B1 *	Q2HJ60
Heterogeneous nuclear ribonucleoprotein F	−1.03	HNRNPF *	Q5E9J1
Proteasome subunit alpha type-4	−1.04	PSMA4 *	Q3ZCK9
26S proteasome regulatory subunit 6B	−1.63	PSMC4 *	Q3T030
Serpin A3-2	−1.36	SERPINA3 *	A2I7M9
14-3-3 protein gamma	−1.02	YWHAG *	A7Z057
Integrin alpha-V		ITGAV	P80746
Integrin beta-1		ITGB1	P53712
Integrin alpha-5		ITGA5	F1MK44
Integrin beta		ITGB3	F1MTN1

**Table 3 ijms-22-05638-t003:** Top early pregnancy-related proteins differentially abundant between BBT-18- and BBT-9-soluble proteins detected by iTRAQ-LC-MS/MS analysis.

Description	Mean Fold Change BBT-18:BBT-9	Gene Name	Accession Number	Reproductive Event
Pregnancy-associated glycoprotein 2	53.54	PAG2	Q28057	co. marker peri-elongation
Secreted seminal-vesicle Ly-6 protein 1	16.24	SSLP1	P83107	co. marker peri-gastrulation
14-3-3 protein sigma	15.76	SFN	Q0VC36	Trophoblast marker
Interferon tau-1	13.13	IFNT1	P15696	Implantation/Co. secretion
Trophoblast Kunitz domain protein 1	10.96	TKDP	Q28201	Trophoblast marker/Co marker peri-elongation
Keratin type II cytoskeletal 8	9.67	KRT8	F1MU12	Trophoblast marker
Ezrin	7.52	EZR	P31976	Implantation
Galectin	5.25	LGALS3	A6QLZ0	co. marker peri-elongation
Interferon tau-3	4.99	IFNT3	P56831	Implantation/Co. secretion
Furin	3.53	FURIN	Q28193	co. marker peri-gastrulation
CD44 antigen	2.91	CD44	Q29423	co. marker peri-gastrulation
Retinoic acid receptor responder 1	2.32	RARRES2	Q29RS5	Trophoblast marker
Macrophage-capping protein	2.24	CAPG, AFCP, MCP	Q865V6	co. marker peri-elongation
Vimentin	−10	VIM	P48616	co. marker peri-gastrulation
72 kDa type IV collagenase	−10	MMP2	F1MKH8	Implantation/Co. secretion
Follistatin-related protein 1	−5.3	FSTL1	Q58D84	Implantation
Fibronectin	−5	FN1	G5E5A9	co. marker peri-gastrulation
Heat shock cognate 71 kDa protein	−3	HSPA8	P19120	Trophoblast marker/Co marker peri-elongation
10 kDa heat shock protein mitochondrial	−2.7	HSPE1	A0A3Q1N8Q5	co. marker peri-elongation
Plastin-3	−2.3	PLS1	A6H742	Trophoblast marker
Fibroblast growth factor receptor	−1.8	FGFR1	F1MQI5	co. marker peri-elongation

**Table 4 ijms-22-05638-t004:** Angiogenic proteins expressed in the soluble fraction and EV-cargo from BBT-secretome, detected by iTRAQ-LC-MS/MS analysis.

Angiogenesis Pathway	Mean Fold Change BBT-18:BBT-9	Gene	Accession Number	Secretome
Catenin Beta-1	co-expressed	CTNNB1	Q0VCX4	EV and soluble proteins
Mitogen-activated protein kinase 1	co-expressed	MAPK1	P46196	EV and soluble proteins
Serine/threonine-protein kinase PAK-1	co-expressed	PAK 1	Q08E52	Soluble proteins
Transforming protein RHOA	co-expressed	RHOA	P61585	EV
MAP2K1	co-expressed	MAP2K1	Q0VD16	EV
Serine/threonine-protein kinase A-RAF	2.7	ARAF	A0A452DI16	Soluble proteins BBT-18
Heat Shock Protein Beta-1	2.29	HSPB1	Q3T149	EV BBT-18
Fibroblast growth factor receptor 1	−1.9	FGFR1	A0A3Q1LUE0	Soluble proteins BBT-9
RHO GTPase-activating protein 1	−4	ARHGAP1	F6RWK1	Soluble proteins BBT-9
RHO-related GTP-binding protein RHOC	−3.57	RHOC	Q1RMJ6	Soluble proteins BBT-9
Vascular cell-adhesion molecule	−2.04	VCAM1	Q9GKR2	EV-BBT-9

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
