# Peer review of "Embryonic Trophectoderm Secretomics Reveals Chemotactic Migration and Intercellular Communication of Endometrial and Circulating MSCs in Embryonic Implantation"

_ijms, 2021, doi:10.3390/ijms22115638_

Round 1
Reviewer 1 Report
The authors presented the functional transfer of protein-containing secreted molecules in cell linage, between embryonic trophectoderm and maternal MSC and its chemotaxis capacity. Moreover, the present topic is important to other studies related to maternal recognition of pregnancy. Moreover, currently, it is a hot topic in science. The manuscript design was clear and straight to present the hypothesis, test it and describe the results. Finally, I believe it is a study that will collaborate with science and I am pleased to “recommend the acceptance” of the manuscript in the present format for publication.
Author Response
Reviewer 1:
The authors presented the functional transfer of protein-containing secreted molecules in cell linage, between embryonic trophectoderm and maternal MSC and its chemotaxis capacity. Moreover, the present topic is important to other studies related to maternal recognition of pregnancy. Moreover, currently, it is a hot topic in science. The manuscript design was clear and straight to present the hypothesis, test it and describe the results. Finally, I believe it is a study that will collaborate with science and I am pleased to “recommend the acceptance” of the manuscript in the present format for publication.
We thank reviewer expert in the field for their positive remarks.
Reviewer 2 Report
The study deals with embryo-maternal exchanges prior to implantation in the bovine species. To this aim, the authors use in vitro tools derived from their previous published work: trophectoderm primary cultures, endometrial mesenchymal stem cell lines and peripheral MSCs derived from male blood. The whole study is well described, the design appropriate and the results nicely presented.
My questions concern the context of the study. For example, why choosing BBT from hatched blastocysts rather than BBT from elongating embryos? establishing eMSCs from heifers rather than cows? pbMSCs from male rather than female blood? is it to build on "naive" specimens regarding to embryo-maternal crosstalks? have an easier access to starting materials? benefit from "independent" samples easier to combine in any new experimental design? Even though previous papers are cited, presenting briefly the whole approach to the readers would make it easier to appreciate all its outcomes. The place to do so might be the introduction, first section of the results or discussion according to authors choice.
Author Response
We thank reviewer expert in the field for positive remarks and constructive comments to improve the description of our work.
Reviewer 2:
The study deals with embryo-maternal exchanges prior to implantation in the bovine species. To this aim, the authors use in vitro tools derived from their previous published work: trophectoderm primary cultures, endometrial mesenchymal stem cell lines and peripheral MSCs derived from male blood. The whole study is well described, the design appropriate and the results nicely presented.
My questions concern the context of the study. For example, why choosing BBT from hatched blastocysts rather than BBT from elongating embryos? establishing eMSCs from heifers rather than cows? pbMSCs from male rather than female blood? is it to build on "naive" specimens regarding to embryo-maternal crosstalks? have an easier access to starting materials? benefit from "independent" samples easier to combine in any new experimental design? Even though previous papers are cited, presenting briefly the whole approach to the readers would make it easier to appreciate all its outcomes. The place to do so might be the introduction, first section of the results or discussion according to authors choice.
Following the reviewer's suggestion, a paragraph has been included in the discussion section (line 348).